# Identification of Suitable Sites Using GIS for Rainwater Harvesting Structures to Meet Irrigation Demand

**Preeti Preeti** [1] , **Yuri Shendryk** [2] **and Ataur Rahman** [1,*]

1   Building XB, School of Engineering, Design and Built Environment, Western Sydney University, Penrith, NSW 1797, Australia
2   Dendra Systems Ltd. Australia, Singleton, NSW 2330, Australia
*   Correspondence: a.rahman@westernsydney.edu.au

**Abstract:** This study uses a multi-criteria decision analysis approach based on geographic information system (GIS) to identify suitable sites for rainwater harvesting (RWH) structures (such as farm dam, check dam and contour bund) to meet irrigation demand in Greater Western Sydney region, New South Wales, Australia. Data on satellite image, soil, climate, and digital elevation model (DEM) were stored in GIS layers and merged to create a ranking system, which were then used to identify suitable RWH (rainwater harvesting) areas. The resulting thematic layers (such as rainfall, land use/land cover, soil type, slope, runoff depth, drainage density, stream order and distance from road) were combined into one overlay to produce map of RWH suitability. The results showed that 9% of the study region is 'very highly suitable' and 25% is 'highly suitable'. On the other hand, 36% of the area, distributed in the north-west, west and south-west of the study region, is 'moderately suitable'. While 21% of the region, distributed in east and south-east part of the region, has 'low suitability' and 9% is found as 'unsuitable area'. The findings of this research will contribute towards wider adoption of RWH in Greater Western Sydney region to meet irrigation demand. The developed methodology can be adapted to any other region/country.

**Keywords:** GIS; multi-criteria decision analysis; rainwater harvesting; site suitability; Sydney





## 1. Introduction

Water is one of the most precious natural resources, which is needed for both human and animal existence as well as for economic growth and development [1,2]. Freshwater shortage has emerged as a critical issue in sustainable development, with communities facing water scarcity issues not just in agriculture and industry, but also in meeting domestic water demands [3,4]. Climate change, population growth, fast industrialization and excessive use of groundwater are considered as the five primary worldwide causes that are putting pressure on safe water supply across the world. As a result, careful management of freshwater resources is essential [5,6]. On the other hand, researchers throughout the world have utilised a number of alternative measures to address the issue of water scarcity [7]. RWH has become a common practice in many water-stressed regions and is regarded as one of the most economically and ecologically beneficial water preservation techniques, addressing water scarcity issues while also reducing the risk of flash floods and alleviating groundwater over-extraction problem [8–10].

The use of RWH is critical in boosting water availability and land productivity in dry areas while preserving water resources. The performance and sustainability of the RWH system for irrigation depend significantly on the identification of appropriate RWH locations [11,12]. However, identifying RWH potential locations is difficult due to the need to consider several competing factors such as hydrological (rainfall–runoff relationship), climatic condition (temperature), geographical (land slope), soil parameters (texture, structure and depth) and socio-economic factors (population size, people's priorities and

preferences with RWH and water regulations) [13–16]. Site selection becomes more difficult and tedious when trying to consider all of these issues, especially when a big watershed is involved. The selection of any RWH site for irrigation is a decision-making process that involves analysing large data sets. Environment and water resources data, in general, are geo-information in nature. As a result, extensive GIS use can provide the tools required to ease data combination [17,18]. The four steps of site selection using GIS techniques include choosing a set of specific criteria, determining level of appropriateness of each criterion, selecting sites for water harvesting, and creating suitability maps for the selected sites. The visual representations of these data combined with data processing in GIS can assist in decision making [19,20]. Multi-criteria analysis (MCA) [21–24] has been used in various environmental and water studies.

Various methods and tools were used by different authors to find the appropriate RWH locations for irrigation, for example, Ammar et al. [25] divided the various methods for RWH site selection into four categories: GIS and Remote Sensing (GIS/RS), hydrological modelling (HM) with GIS/RS, MCA incorporated with HM and GIS/RS, and MCA incorporated with GIS. In GIS/RS approach, researchers have integrated both the biophysical and socio-economic parameters for RWH site selection. To improve the accuracy and precision of runoff assessment and suitable site selection, a range of hydrological models have been merged with GIS/RS, namely, soil conservation service-curve number (SCS-CN), watershed modelling system (WMS) and soil and water assessment (SWAT).

For RWH site selection, several studies have been conducted utilising MCA combined with HM and GIS [6,26–29]. All methodologies and applications utilized in previous research studies associated with the potential site selection for RWH have some constraints, however the GIS/RS tool is initial stage application for identifying suitable sites, while the integration of MCA and GIS-based HM are highly recommended methods for more precise outcome. MCA (AHP) combined with GIS has a great potential for RWH site selection and has been used in several studies globally to find potential RWH locations [25,30]. It is a multi-criteria decision-making tool that uses a structured method based on mathematics and expert knowledge to organise and evaluate complicated decisions [31,32]. It is a favourable decision-supporting technique for solving multiplex problems [33]. Along with the GIS platform, it has been identified as the best relevant decision approach for the identification of possible RWH locations even in data-poor region.

There have been limited application of these techniques for a large area such as Western Sydney in Australia. To fill this knowledge gap, this study aims to create a suitability map of possible RWH locations in the Greater Western Sydney (GWS) region of New South Wales, Australia. The final suitability map depicts the spatial distribution of areas suitable for putting RWH structures (such as farm dam, check dam and contour bund) to meet irrigation demand. GWS is selected for the study location due to its strategic location, which is expected to have a high population density in the near future, and as a result, a high water demand. The findings of this research will assist relevant authorities in efficiently planning and implementing water resources management plan, reducing reliance on groundwater, and ensuring long-term water supply for irrigation use in the study area. The remainder of this paper is organized as follows, the study region and data are described in Section 2. The methodology, data processing and layer standardisations are presented in Section 3. The results, conclusions and recommendations for future research are discussed in Sections 4 and 5.

## 2. Study Area and Data

Great Western Sydney (GWS) is situated in the western region of Australia's Sydney metropolitan area, as shown in Figure 1. From Windsor in the north to Campbelltown in the south, and from Parramatta in the east to Penrith and the Blue Mountains in the west, it encompasses 13 local government districts. The selected area, which has one of the fastest increasing populations, is anticipated to have 2.6 million residents living there. GWS spans an area of around 5800 km$^2$. It is Australia's third biggest economy, behind Sydney

and Melbourne. Western Sydney experiences a humid subtropical climate. The first few months of the year, February through April, are wetter, and the months of July through December are drier. Figure 2 shows the locations of the selected rainfall stations, while Table 1 provides the list of selected rainfall stations, coordinates (latitude and longitude), duration and average annual rainfall (AAR) for the selected stations. The daily rainfall data for the historical period at the selected locations were obtained from the Australian Bureau of Meteorology (BOM). The selected sites have mean annual rainfall of 862 mm/year, which ranges from 638 to 1408 mm/year. More details of the used data are provided in Table 2. The study area map of the Greater Sydney region is shown in Figure 1.

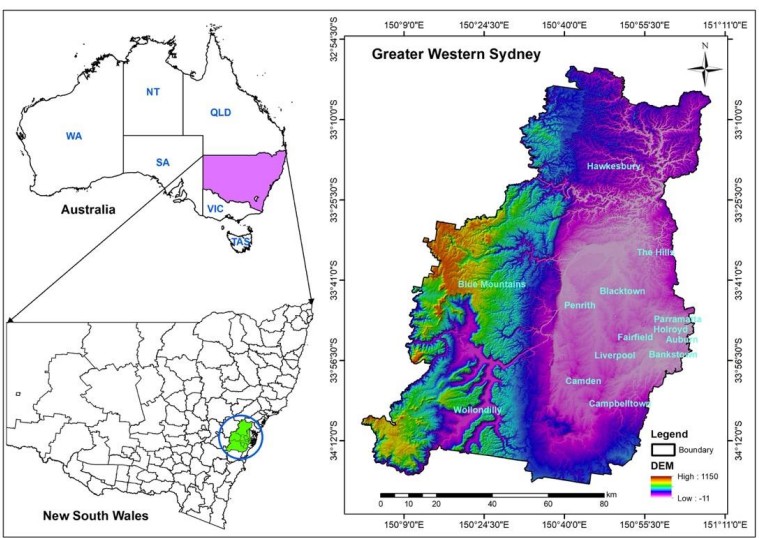

**Figure 1.** Study area map showing Greater Sydney Region.

**Table 1.** Summary of selected rainfall stations data.

| Station Name | Station No. | Lat | Long | Duration | AAR (mm) |
|---|---|---|---|---|---|
| Bankstown Airport AWS | 66,137 | 33.92 | 150.98 | 1969–2021 | 868 |
| Milperra Bridge (Georges River) | 66,168 | 33.93 | 150.98 | 2001–2019 | 638 |
| Seven Hills (Radio FM 103.2) | 67,110 | 33.79 | 150.92 | 2001–2020 | 744 |
| Greystanes (Bathurst Street) | 67,017 | 33.81 | 150.94 | 2002–2021 | 863 |
| Prospect Reservoir | 67,019 | 33.82 | 150.91 | 1887–2021 | 879 |
| Seven Hills (Collin St) | 67,026 | 33.77 | 150.93 | 1950–2020 | 913 |
| Katoomba (Farnells Rd) | 63,039 | 33.71 | 150.3 | 1910–2021 | 1408 |
| Springwood (Valley Heights) | 63,077 | 33.71 | 150.59 | 1883–2020 | 1082 |
| Faulconbridge (St Georges Crescent) | 63,028 | 33.69 | 150.53 | 1937–2021 | 1136 |
| Faulconbridge (Great Western Hwy) | 63,081 | 33.69 | 150.55 | 2002–2021 | 1059 |
| Camden Airport AWS | 68,192 | 34.04 | 150.69 | 1972–2021 | 796 |
| Camden (Brownlow Hill) | 68,007 | 34.03 | 150.65 | 1883–2021 | 744 |
| Cawdor (Woodburn) | 68,122 | 34.1 | 150.64 | 1963–2021 | 770 |
| Mount Annan | 68,254 | 34.06 | 150.77 | 2007–2021 | 720 |
| Campbelltown (Kentlyn (Georges River Roa | 68,160 | 34.05 | 150.88 | 1967–2021 | 765 |
| Mount Annan (Australian Botanic Garden) | 68,254 | 34.07 | 150.77 | 2002–2021 | 747 |
| Horsley Park Equestrian Centre | 67,119 | 33.85 | 150.86 | 1998–2021 | 780 |
| Abbotsbury (Fairfield (City Farm)) | 67,114 | 33.87 | 150.86 | 2001–2021 | 719 |
| Richmond RAAF | 67,105 | 33.6 | 150.78 | 1995–2021 | 741 |
| Richmond—UWS | 67,021 | 33.62 | 150.75 | 1881–2021 | 798 |
| Castlereagh (Castlereagh Road) | 67,002 | 33.67 | 150.67 | 1940–2021 | 840 |
| Sydney Olympic Park AWS (Archery Centre) | 66,212 | 33.83 | 151.07 | 2012–2021 | 1090 |
| North Parramatta (Burnside Homes) | 67,111 | 33.79 | 151.02 | 2001–2021 | 930 |
| Parramatta North | 66,124 | 33.79 | 151.02 | 1966–2021 | 968 |
| Orchard Hills Treatment Works | 67,084 | 33.8 | 150.71 | 1971–2021 | 828 |
| Penrith Lakes AWS | 67,113 | 33.72 | 150.68 | 1996–2021 | 740 |
| Box Hill (Mccall Gardens) | 67,104 | 33.65 | 150.89 | 1991–2021 | 727 |
| Baulkham Hills Eucalyptus Ct | 67,109 | 33.77 | 150.98 | 2001–2021 | 843 |
| Picton Council Depot | 68,052 | 34.17 | 150.61 | 1880–2018 | 801 |
| Cawdor (Woodburn) | 68,122 | 34.1 | 150.64 | 1963–2021 | 769 |
| Douglas Park (St. Marys Towers) | 68,200 | 34.21 | 150.71 | 1975–2021 | 742 |
| Colo Heights (Mountain Pines) | 61,211 | 33.36 | 150.71 | 1963–2021 | 1028 |
| St Albans (Espie St) | 61,217 | 33.29 | 150.97 | 1963–2021 | 937 |
| Oakdale (Cooyong Park) | 68,125 | 34.09 | 150.5 | 1964–2021 | 898 |

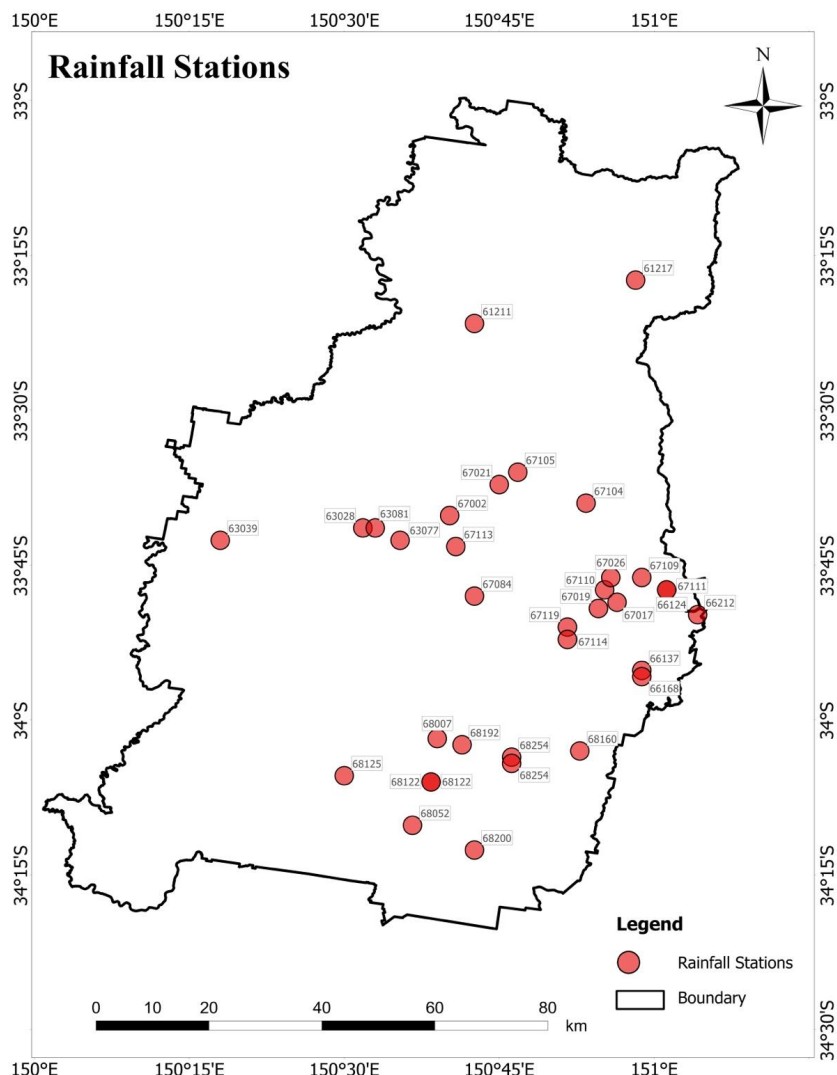

**Figure 2.** Rainfall stations map of the study area.

**Table 2.** Summary of source of data used in this study.

| Data Type | Source | Year | Description |
|---|---|---|---|
| Sentinel-2 | Esri | 2020 | Land use/Land cover |
| Shuttle Radar Topographic Mission (SRTM 30 m) | Geoscience Australia | 2000 | Digital elevation model (DEM) generated from SRTM, UTM—WGS84, Zone 56 S |
| Climate Data | Bureau of Meteorology, Australia | 1880–2019 | Monthly and daily climate data |

The location map of the selected rainfall stations of the study area is shown in Figure 2, while Table 2 provides the summary of data sources used in this study.

## 3. Methodology

This study adopted a GIS based muti-criteria decision analysis (MCDA) modelling approach to standardise various data layers. For locating an ideal RWH site for irrigation, the thematic layers must have certain feature such as (l) kind of soil having sufficient clay to retain water and limit seepage; (2) a flat terrain that allows for economic construction; and (3) adequate amount of runoff that can be stored on depressed land or in rainwater tanks [34]. The weighted linear combination (WLC) was selected to rank and prioritize

the thematic layers dataset. The thematic layers including land use/land cover, soil type, slope, stream order, drainage density, and the runoff layers were prepared, classified, and integrated into GIS environment. Furthermore, a suitability rating was also allocated to each layer in this analysis, and the layers were then combined using the ArcGIS program. Afterwards, the various layers that had been weighted were combined to create the final suitability map. Figure 3 shows the main steps used in site selection of RWH structures.

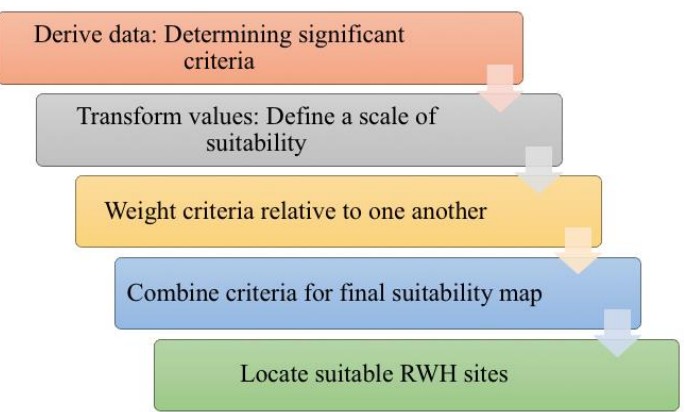

**Figure 3.** Main steps used in RWH site suitability data processing.

Normalization, buffering, and weighting tasks within data processing included the following steps: (i) reclassifying or normalising biophysical criteria maps based by grading the parameters on a single scale, (ii) reclassifying socioeconomic standards by applying buffer zones for each of the criteria, and (iii) calculating relative importance weights (RIW) based on expert evaluations that would provide the final value of every criterion for RWH system. Figure 4 shows a flow diagram of the methodology adopted to produce the final RWH potential map.

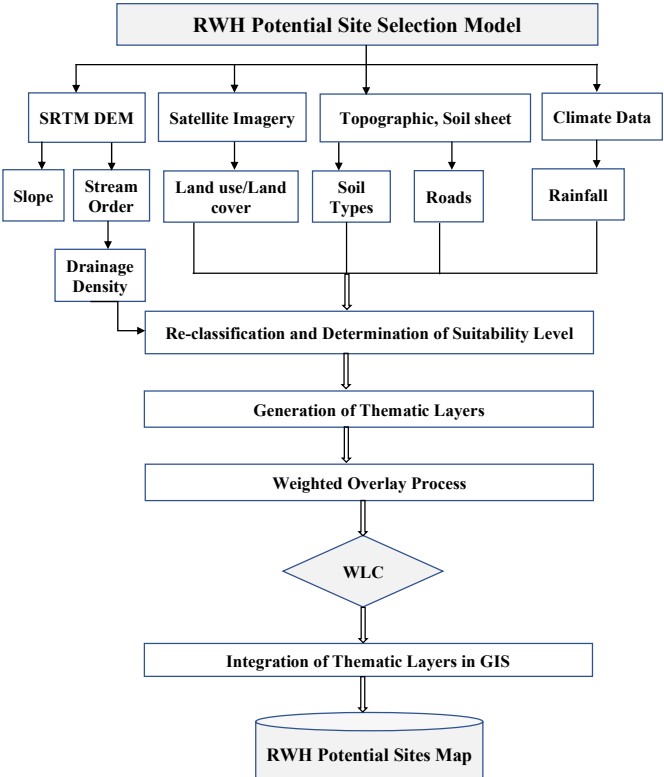

**Figure 4.** Conceptual methodology flowchart used in this study.

### 3.1. Digital Elevation Model (DEM)

A DEM offers digital illustration of some of earth's landscape over two-dimensional surface. DEMs are frequently made using a regular array of elevation data from geographic maps and aerophoto [35]. The NASA shuttle radar topographic mission (SRTM) used inter­ferometric sythetic aperture radar (InSAR) to acquire elevation data at a spatial resolution of 30 m [36]. The DEM gives crucial information about a basin's topographical features, including slope, aspect, perspective three-dimensional view, hill shading, flow direction, flow accumulation, stream order etc. The D8 algorithm is a useful method to interpret the elevation data and extract the above-mentioned features. The DEM map of the study area is shown in Figure 5.

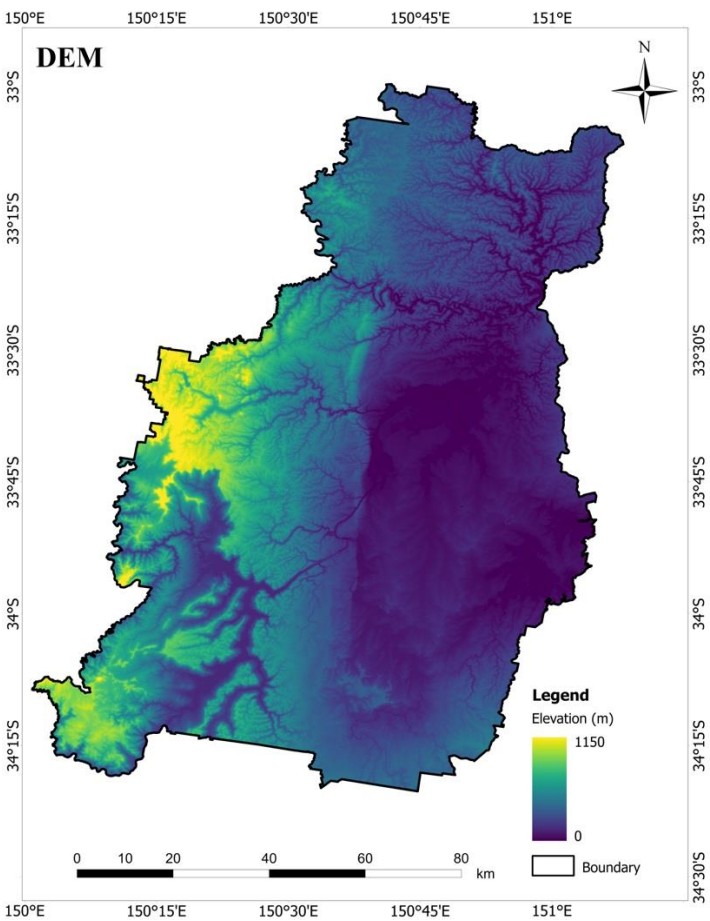

**Figure 5.** The digital elevation model (DEM) of the study area.

### 3.2. Slope

The slope is an important parameter to consider when choosing a location for a RWH structure since it affects the size of the harvesting structure and the amount of water stored. If a large volume of water is needed to meet irrigation demand, high slope areas may not be suitable as the storage structure (such as farm dam) could be more expensive in such cases. Areas with a medium to low slope are more practical to reduce capital cost of storage structure [37]. The spatial analyst tool capabilities of the ArcGIS programme were used to process the resulting DEM. The slope tool in the surface group of the spatial analyst tools in the ArcGIS environment was used to create a slope map. The generated slopes of the study area were divided into percentage categories in terms of runoff generation. According to Critchley et al. [38], areas with a slope greater than 5% are not advised for RWH, because of the uneven distribution of runoff and financial loss resulted by the extensive earthwork needed.

### 3.3. Stream Order

The connection of the tributaries determines a stream's order. A lower stream order generally denotes more porosity and infiltration, which is important for its examination while choosing a prospective RWH location [39]. In this study, the spatial analysis tools are used to extract and create the stream order map using DEM data and a stream order layer value ranging from 1 to 6. According to AL-Ardeeni [40], the criteria for the stream order must be greater than the third order. Figure 6 shows the key steps used to extract the stream order from DEM.

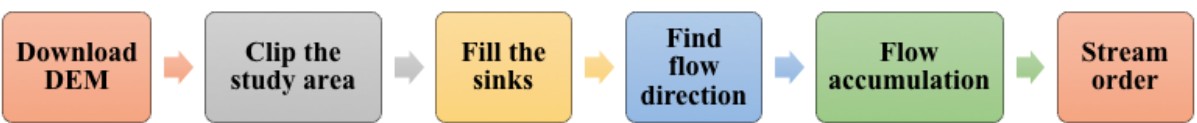

**Figure 6.** Key steps used to extract the stream order from DEM.

### 3.4. Drainage Density

The movement of groundwater, recharge, and hydrogeological processes are all fundamentally influenced by drainage density. The volume of runoff loss via infiltration depends on drainage density and the lower the drainage density, the lower the RWH potential [39]. Using the DEM in the GIS environment, the drainage density of the study region was calculated. The drainage density is defined as the total stream length per unit catchment area as shown by Equation (1) [41]:

$$DD = \frac{\sum_{i=1}^{n} L}{A} \tag{1}$$

where DD is the drainage density, n is the number of streams, L is the stream length (km), and A is the drainage area (km$^2$).

### 3.5. Land Use/Land Cover (LULC)

Any watershed's land use pattern affects runoff, infiltration and evapotranspiration [42]. LULC maps are needed in order to compute hydrological components more precisely. Images can be created using image-processing techniques that show specific features, most notably the different types of land use and cover, such as regions with vegetation, water bodies and barren soils. Vegetation cover affects the water harvesting land suitability. The hydrological response of the watershed is significantly influenced by the LULC pattern and precipitation [43]. The study area's LULC map was divided into eight major classes, including waterbodies, built-up area, trees, grass, crops, flooded vegetation, shrub and bare ground. The land-use/land-cover pattern was categorized under five levels of suitability: very high suitability, high suitability, medium suitability, low suitability and unsuitable classes.

### 3.6. Soil

The soil map of any watershed is essential for understanding the land use and geography of an area of interest. When choosing a RWH location to supply water for irrigation, the type of soil is of critical consideration [44]. The soil's permeability is main parameter in determining the rate of infiltration and the amount of water stored in the soil layers. The proportions of sand, silt, and clay are used to identify the soil textural class. Since they can store more water, the fine and medium grained soils are more suited for collecting rainwater. The study area's soil map showed a variety of soil types, including alluvial, lithosols, brown earth, podzols and red podzolic soils. Following this, all these classes are categorized under five levels of suitability: very high suitability, high suitability, medium suitability, low suitability and unsuitable classes.

*3.7. Rainfall*

Australia is a large continent, and the amount, timing, and distribution of rainfall vary greatly from place to place. When choosing a suitable site for RWH, rainfall is the most important and receives the highest weighted factor. The distribution map of the average yearly rainfall for given area is created using inverse distance weight spatial interpolation technique. The RWH site selection depends on the rainfall availability that is gathered in a particular location [3]. In addition, one of the primary sources of water loss that affects the RWH process is ground evaporation, i.e., a low evaporation rate is a sign that a region is suitable for water harvesting [37]. The rainfall variability of the area ranges 644–1408 mm and the criteria of rainfall were selected greater than 600 mm. Four levels of appropriateness were used to characterise the distribution of rainfall values in the study area: very high suitability, high suitability, medium suitability, and unsuitable classes.

*3.8. Roads*

Spatial data techniques, such as buffering, union, and raster reclassification were chosen to evaluate socioeconomic criteria such as distance to roads and settlements. Although socioeconomic status is not directly related to the proximity of roads and settlements, in rural areas, developments tend to happen near road network. The distance to roads was calculated using the Euclidean distance function. The Euclidean distance values ranged 0 to 9452 m. It should be noted that the harvested rainwater should be used locally for irrigation to avoid high water transportation cost.

*3.9. RWH Potential Map using Weighted Linear Combination (WLC)*

The WLC approach is flexible in locating the suitable sites. This method was previously used in a number of studies [45–47] and it was found to be effective in combining the RIW and final standardized/reclassified maps to produce a final suitability map for each RWH system. As a result, the WLC approach was applied in this study using the ArcGIS software after determining the normal weight of each layer and sub-layer. Based upon findings and evaluations of previous studies, the following weight values were used in this study: Rainfall: 20; Soil: 20, LULC: 20; Drainage Density: 20, Slope: 10, Stream order: 5, and Distance to Roads and Settlement: 5. Equation (2) was used to calculate the WLC suitability maps:

$$S = \sum RIW_i * SL_i \tag{2}$$

where: S is the suitable site, $RIW_i$ is the relative importance weight of the input layer i (rainfall, slope, stream order, drainage density, land use/land cover and soil type) for RWH system and $SL_i$ is the degree of appropriateness/suitability of the input layer. The higher the $SL_i$ of a given cell is, the better suited it is for RWH system. Equation (2) produced a dimensionless value called S. Five categories were used to grade the S: unsuitability, low suitability, medium suitability, high suitability, and very high suitability. This approach was used in several earlier studies [6,13,28,46,48]. The primary thresholds used to determine the site appropriateness levels for the RWH systems are the final weights of every standard, preferred values for the criterion classes, and buffering zones. The final RWH suitability map is created by combining this layer with the WLC method's output layer.

**4. Results and Discussion**

The criterion layers must be on the same scale and in the same units for the WLC to apply. Further categorisation requires the conversion of vector layers such as roads, soil, and stream order into raster format, as shown below.

*4.1. Rainfall*

Figure 7 depicts the average annual rainfall distribution and reclassified map of the study area. Rainfall map shows how rainfall amounts are distributed over the study area; however, this does not always imply that potential harvesting is concentrated in areas with high rainfall. Low values occupy the central eastern and southeast parts of the study area.

However, high values are observed in the central western and southwestern borders. The rainfall map of the study area is shown in Figure 7.

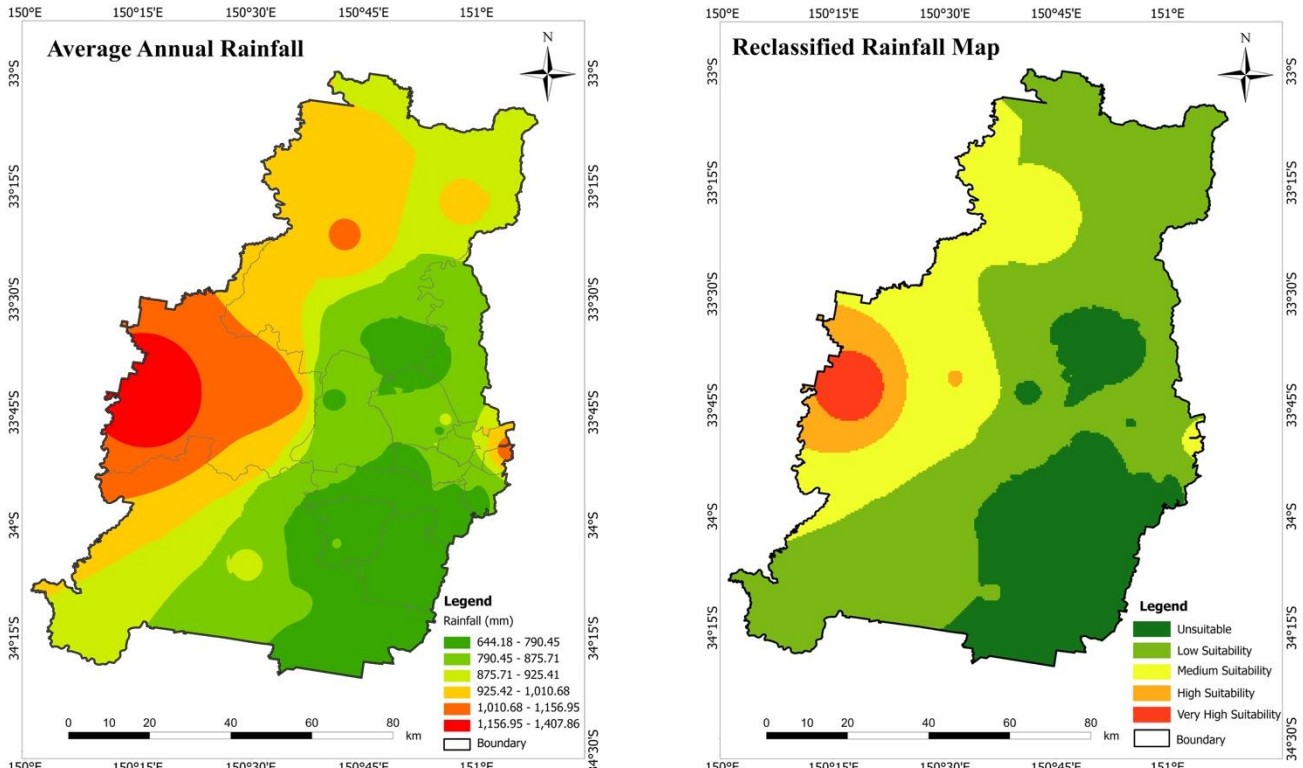

**Figure 7.** Rainfall map of the study area average annual rainfall (**left**); reclassified rainfall (**right**).

### 4.2. Slope

The study area's slope and reclassified suitability map are shown in Figure 8. ArcGIS software was used to create a slope map using the SRTM DEM. The SRTM DEM was used to produce a slope map with the GIS environment. The study area has a wide range of slopes, from mild to quite high. Five categories of slope percentage are distinguished: flat (0–5%), mild (from 5–10%), moderate (10–15%), steep (15–30%), and mountainous (>30%). Slope has an impact on surface water runoff, recharge, and flow, and is one of the most important factors to consider when choosing and implementing RWH structures. A total of 25.63% of the study area is classified as having a hilly slope, which is least suited area for RWH. Additionally, the coverage areas for mild and moderate slopes are 16% and 9%, respectively, as shown in Figure 8. A slope of less than 5% is ideal as suggested by Adham et al. [13]. Therefore, 25.74% of the research area is extremely suited for RWH, as shown by the slope map (Figure 8). Table 3 shows distribution of slope suitability classes.

**Table 3.** Distribution of slope suitability classes.

| Factor | Interval | Rate | Suitability | Coverage (km²) | Coverage (%) |
|---|---|---|---|---|---|
| Slope (%) | >30 | 1 | Unsuitable | 2402.9 | 25.63 |
| | 15–30 | 2 | Low Suitability | 2113.12 | 22.54 |
| | 10–15 | 3 | Medium Suitability | 929.61 | 9.91 |
| | 5–10 | 4 | High Suitability | 1516.69 | 16.18 |
| | <5 | 5 | Very High Suitability | 2413.82 | 25.74 |

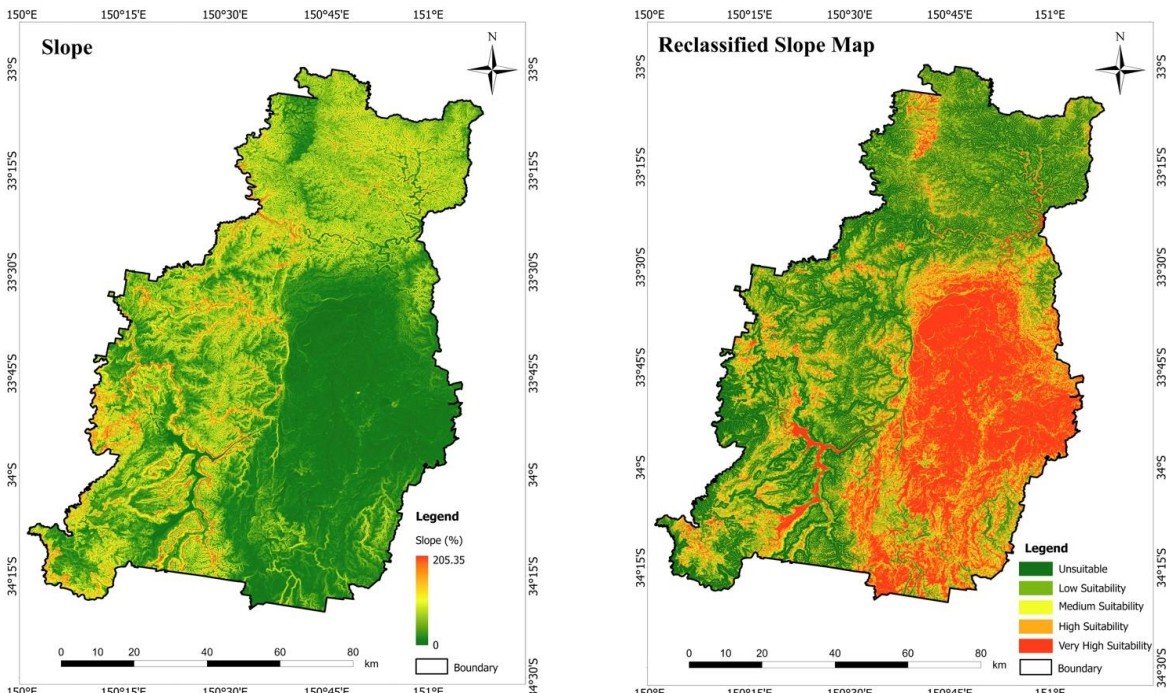

**Figure 8.** Study area's slope and reclassified slope map slope (**left**); reclassified slope (**right**).

### 4.3. Stream Order

Figure 9 depicts the stream order and reclassified map of the study area. As shown below, stream order layer values range 1 to 6. According to Ammar et al. [25], a stream order of less than 3 is not appropriate for water harvesting. A higher stream order is thought to have better RWH potential.

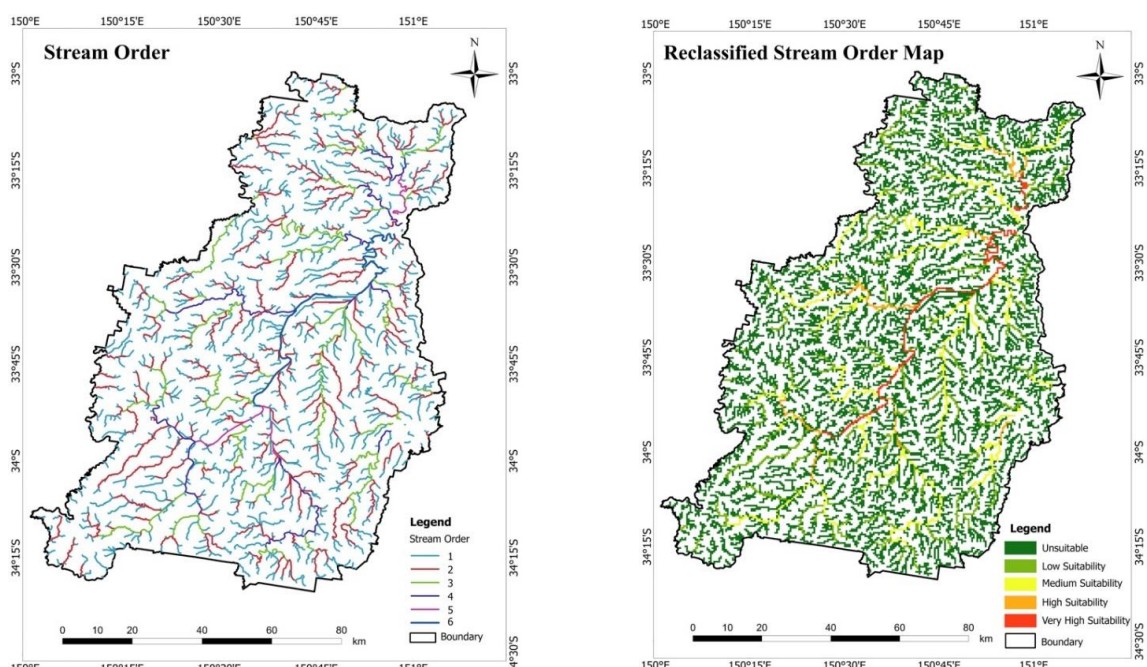

**Figure 9.** Study area's stream order and reclassified map stream order (**left**); reclassified stream order (**right**).

### 4.4. Drainage Density

Figure 10 depicts the drainage density and reclassification map of the study area, and Table 4 displays the drainage density suitability class table. RWH is more likely to occur in areas with higher drainage densities. According to the map in Figure 10, the study area's 80% has the highest concentration of stream densities.

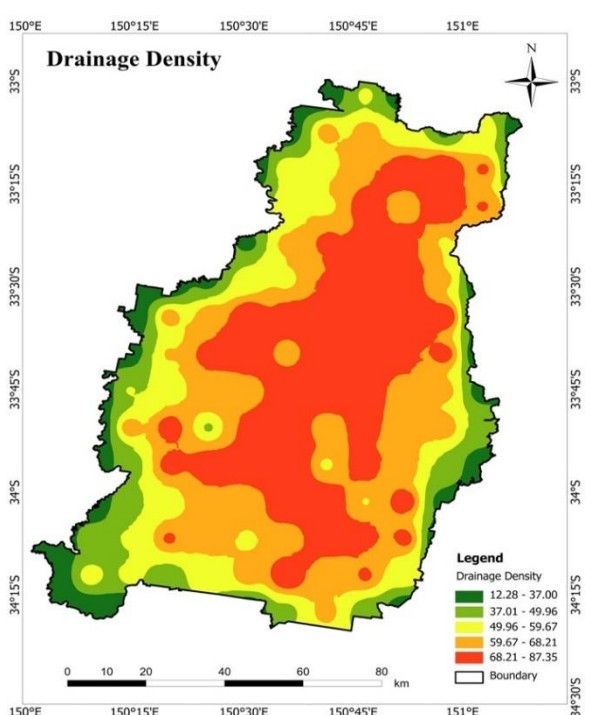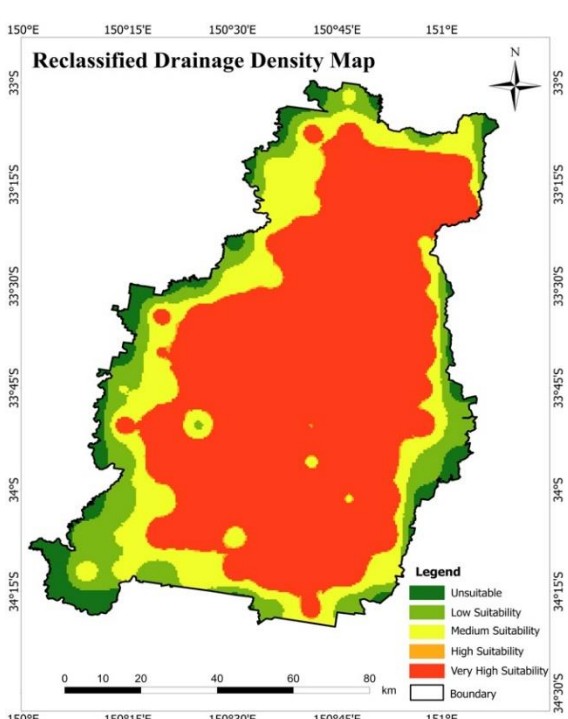

**Figure 10.** Study area's drainage density map drainage density (**left**); reclassified drainage density (**right**).

**Table 4.** Distribution of drainage density suitability classes.

| Factor | Interval | Rate | Suitability | Coverage (km²) | Coverage (%) |
|---|---|---|---|---|---|
| | <30 | 1 | Unsuitable | 2.61 | 0.03 |
| Drainage | 30–40 | 2 | Low Suitability | 275.39 | 2.93 |
| Density | 40–50 | 3 | Medium Suitability | 1448.65 | 15.41 |
| (km²) | 50–60 | 4 | High Suitability | 7589.08 | 80.74 |
| | 60–87.35 | 5 | Very High Suitability | 83.9 | 0.89 |

### 4.5. LULC

Figure 11 depicts the study area's LULC map and identifies the seven main categories namely waterbodies, built-up, scrub/shrub, trees and grass (mixed forest), crops, flooded vegetation and barren land. All of these classes are then divided into five primary classes based upon their suitability as shown in the reclassification map. Major portion of the study area (about 56%) is mixed forest followed by scrub (23.22%), bare ground (2.14%) and waterbodies and built-up (17.98%) areas. For water harvesting zones and structures, it is often advised to employ land-use types such as bare ground and sparsely vegetated land. Table 5 provides the LULC suitability class. Conversely, water bodies and settlements occupying about 17.98% are considered to be unsuitable for harvesting rainwater. Thus, it is clear that both scrub and forest, which take up 79% of the study region are the most favourable areas for collecting rainwater for irrigation use.

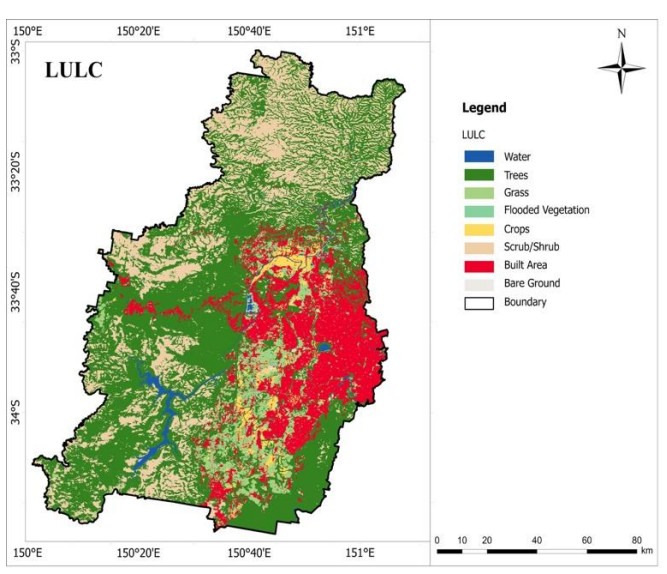
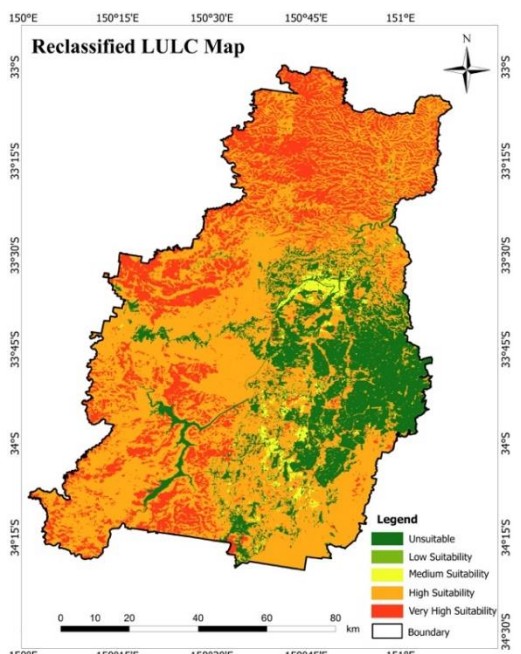

**Figure 11.** Land use/land cover and reclassified suitability map LULC (**left**); reclassified LULC (**right**).

**Table 5.** Distribution of land use/ land cover suitability classes.

| Factor | Type | Rate | Suitability | Coverage (km²) | Coverage (%) |
|---|---|---|---|---|---|
| | Water Body/Built Area | 1 | Unsuitable | 1689.58 | 17.98 |
| Land use/Land cover | Flooded Vegetation | 2 | Low Suitability | 5.68 | 0.06 |
| | Bare Ground | 3 | Medium Suitability | 201.09 | 2.14 |
| | Grass/Trees | 4 | High Suitability | 5319.63 | 56.61 |
| | Scrub/Shrub | 5 | Very High Suitability | 2181.83 | 23.22 |

*4.6. Soil*

Figure 12 depicts the study area's soil type and reclassified map based upon their suitability. ArcGIS was used to digitize the soil type map and a final reclassified raster map was generated. The selected area's soil map shows a variety of soil types such as alluvial, earthy sands, lithosols, brown earth, podzols and red podzolic soils. All of these classes are then categorized into five classes of suitability: very high suitability (clay), high suitability (silt clay), medium suitability (sandy clay), low suitability (sandy loam) and unsuitable (others) class as shown in Table 6. The major portion of the study area is sandy clay (about 49%) followed by sandy loam (27.45%), silt clay (19.46%) and clay (2.58%).

**Table 6.** Distribution of soil types of suitability classes.

| Factor | Type | Rate | Suitability | Coverage (km²) | Coverage (%) |
|---|---|---|---|---|---|
| | Others | 1 | Unsuitable | 149.54 | 1.6 |
| | Sandy Loam | 2 | Low Suitability | 2571.06 | 27.45 |
| Soil Type | Sandy Clay | 3 | Medium Suitability | 4582.66 | 48.92 |
| | Silty Clay | 4 | High Suitability | 1822.72 | 19.46 |
| | Clay | 5 | Very High Suitability | 241.26 | 2.58 |

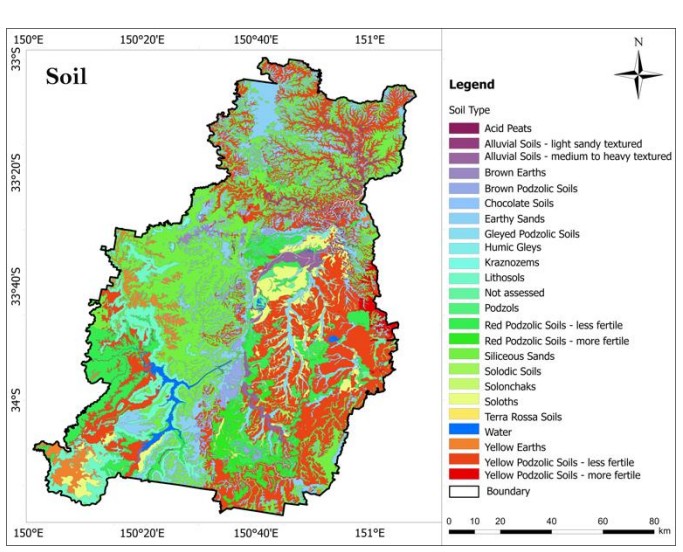

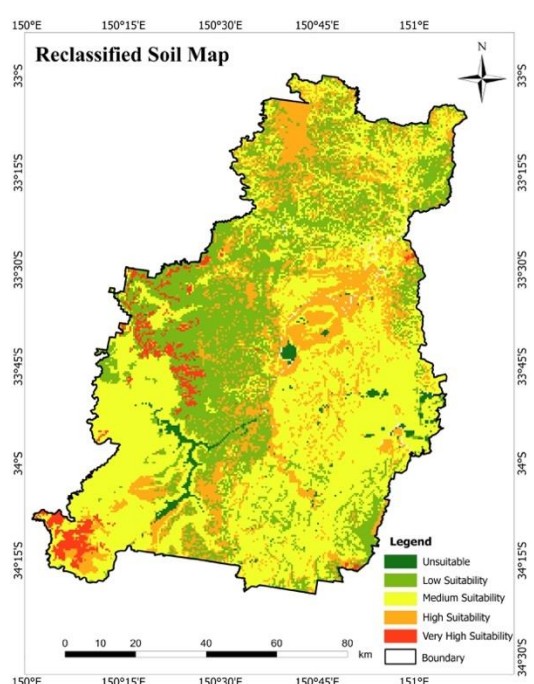

**Figure 12.** Soil types and reclassified suitability map soil (**left**); reclassified soil (**right**).

### 4.7. Distance to Roads

Euclidean distance function, as illustrated in Figure 13, is used to calculate the distance to roadways, with values ranging from 0 to 9452 m. In Figure 13, a reclassified map with suitability classifications is also displayed.

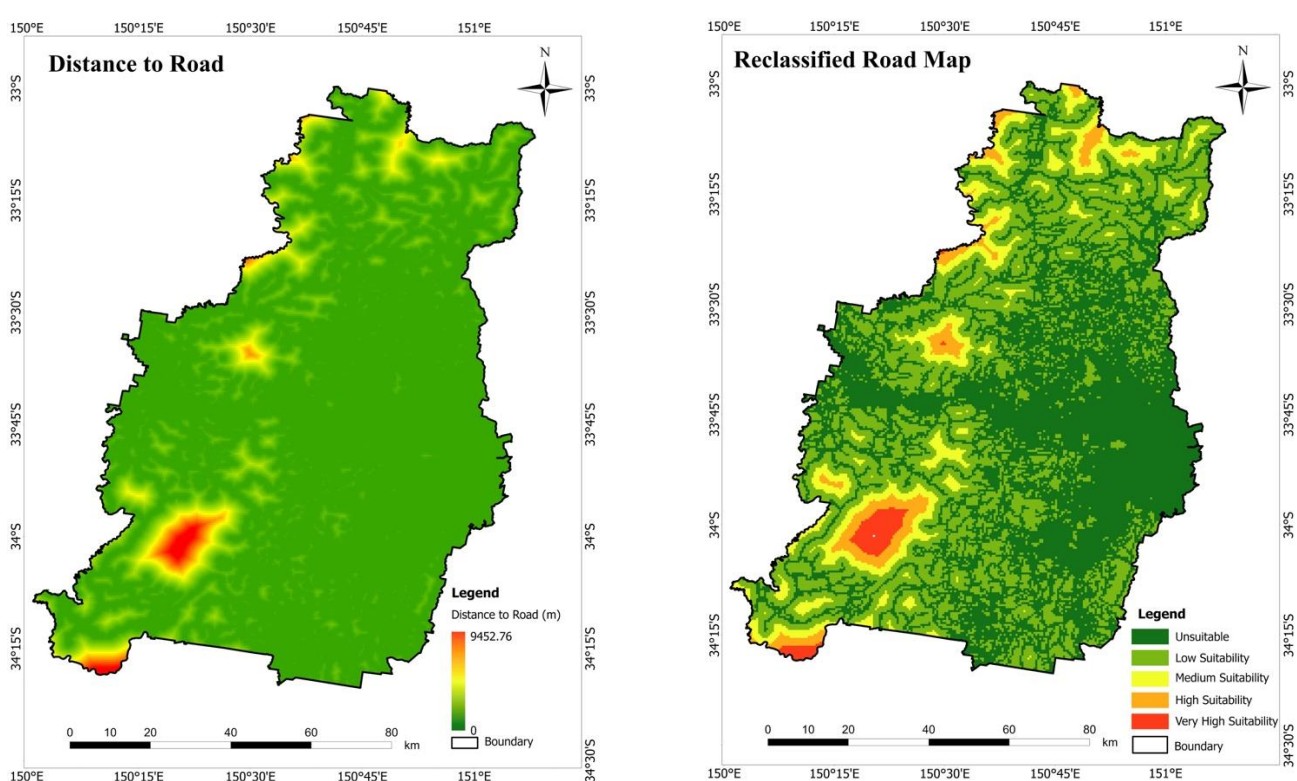

**Figure 13.** Distance to road map of the study area distance to road (**left**); reclassified distance to road (**right**).

*4.8. Final Suitability Map*

The selected areas supporting RWH structure were divided in sub-areas based upon their suitability. In order to do this, the suitability classes were categorized into areas based on ranking that were, very high suitability, high suitability, medium suitability, low suitability and unsuitable regions, as illustrated in Figure 14. The suitability rankings for these locations range from 5 to 1, with 5 denoting the most appropriate sites and 1 denoting unsuitable/restricted sites. According to Figure 15, the study area was most heavily represented by the moderately appropriate regions, which made up 36% of the overall area. The regions with a 9% regional coverage were at the lowest scale and were entirely disregarded as unsuitable. About 8% and 25%, respectively, were covered in the high and very high appropriate zones. The majority of the regions were given the greatest attention for identifying the RWH structures, as they were best-suited sites for these areas in terms of rainfall, slope, stream order, drainage density, soil, land use/cover and distance to roads. Figure 14 shows the final RWH sites suitablility map of the study area while Figure 15 shows the distribution of areas covered by the RWH suitability classes.

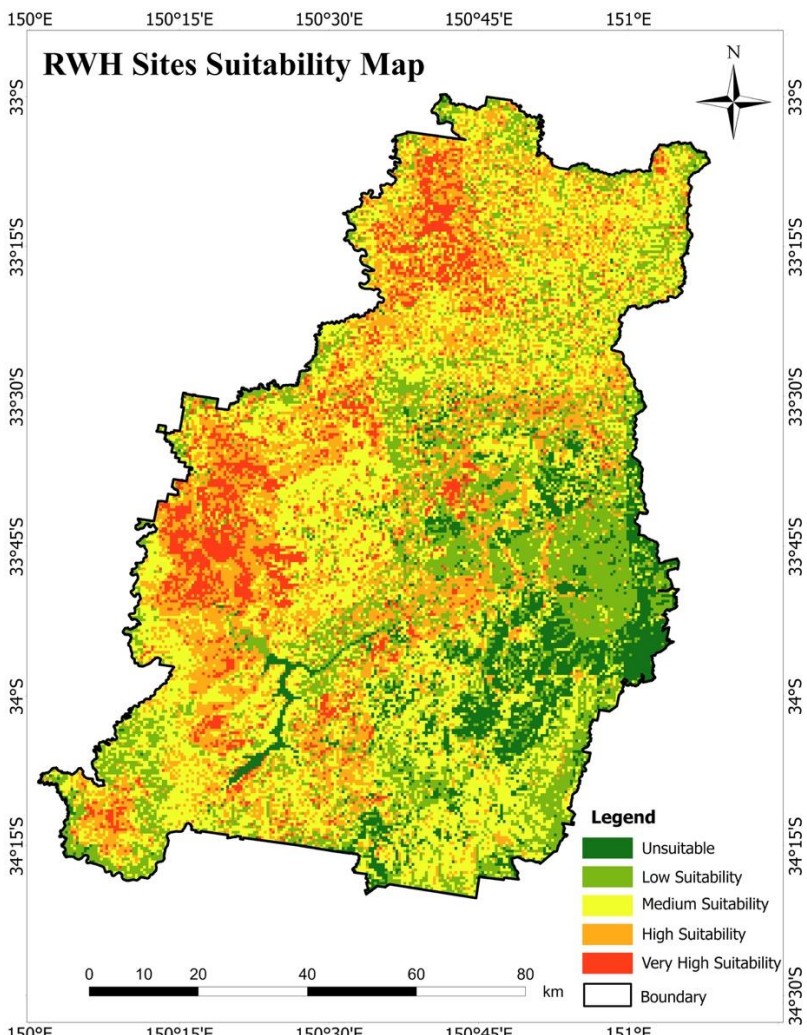

**Figure 14.** RWH final suitability map of the study area.

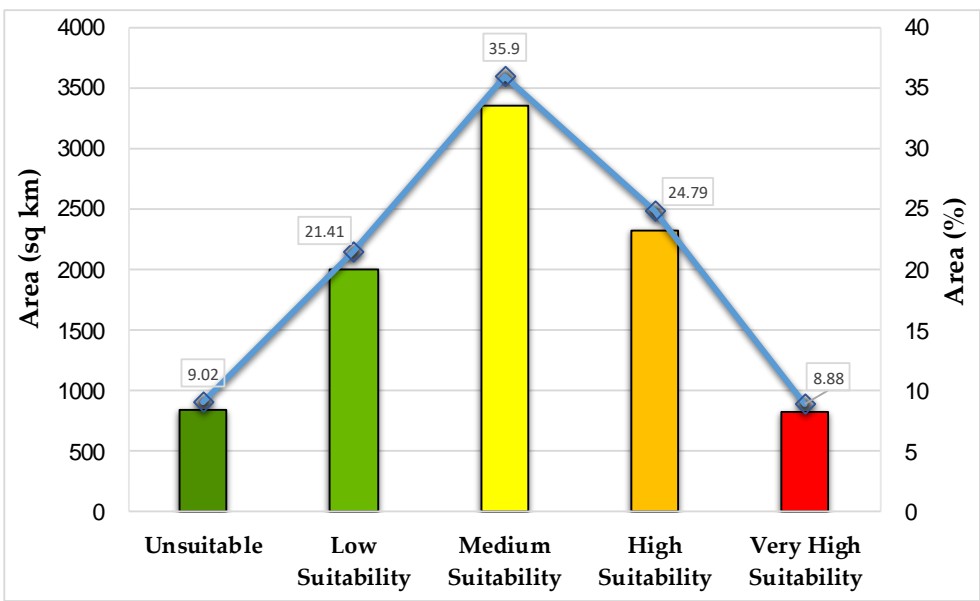

**Figure 15.** Distribution of areas covered by the RWH suitability classes.

## 5. Conclusions

RWH is a viable method for managing water deficiency issues effectively by increasing water availability over the longer term. The current study used the GIS-based MCDA technique to address this issue and develop a reliable methodology for finding potential locations for putting RWH structures for irrigation use. In this study, ArcGIS was found a convenient tool for combining data from various thematic layers to identify appropriate locations. Finding optimal locations for rainwater collection and storage requires a versatile, efficient, and comprehensive data source and tool such as ArcGIS. The study considered various factors into account, including slope, stream order, drainage density, land use/cover, soil types and distance to roads. According to the findings, 25% and 9% of the study area, respectively, are highly appropriate for RWH. On the other hand, the remaining 36% distributed in the north-west, west and south-west is moderately suitable for the collection of rainwater. About 9% of the area is either restricted or unsuitable for water harvesting, with a low suitability rate of 21% distributed over the east and south-east.

The findings of this study will facilitate Australian water authorities and decision-makers in strengthening future development strategies for managing water scarcity within the selected study area. The developed method for figuring out appropriate RWH sites for irrigation use requires minimal work and may be applied to other water-scarce areas. However, further research is necessary before implementing RWH system, including complete site characterizations, socio-economic activity analyses, and in-depth audit of the planned RWH locations.

**Author Contributions:** P.P. contributed to data analysis, methodology, first draft, A.R. proposed the conceptual framework of the paper and supervised the analysis and improved writing, Y.S. checked the results and enhanced the writing of the paper. All authors have read and agreed to the published version of the manuscript.

**Funding:** This research received no external funding.

**Data Availability Statement:** The rainfall data used can be obtained from Australian Bureau of Meteorology by paying a fee.

**Acknowledgments:** We acknowledge the support from the Australian Bureau of Meteorology for providing rainfall data for this study.

**Conflicts of Interest:** The authors declare that there is no conflict of interest.

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
