# Peer review of "Identification of Suitable Sites Using GIS for Rainwater Harvesting Structures to Meet Irrigation Demand"

_water, doi:10.3390/w14213480_

Round 1

Reviewer 1 Report

The manuscript is interesting and deals with an important topic, especially in the face of increasing climate change. The editorial staff of the article could have been more careful. The necessary fixes are listed below:

1.       The paper is more straightforward to read if each table and figure are immediately preceded by a reference in the text. If possible, please include the designation in the text before the table and the drawing. I mean Figure 1, Figure 2, Figure 7, Figure 14, Figure 15, Table 1 and Table 2.

2.       If possible, please move the name of table 1 (line 118) to the next page.

3.       Does the table formatting comply with the publisher's recommendations? Please, check the Instructions for Authors and use the Microsoft Word template.

4.       Please describe in points the tasks listed in lines 140-146. It will be much more readable.

5.       There is no reference in the text to  Figure 6.

6.       If you describe the parameter designations (equation 1), use a colon after the "where" (line 197).

7.       Do not use italic type in equation 2 (line 252) if equation 1 does not have italics (line 195).

8.       If possible, move the description of Figure 10 so that it is on the same page as the Figure 10.

9.       If possible, move the description of Table 5 so that it is on the same page as the Table 5.

10.   In Figure 12, the name "Soli" should not contain capital letters. In the remaining drawings, only the first letter of the description is capital.

11.   If possible, move the name of section 5.7 to the next page (line 347).

1    12.   Please add horizontal extension lines in Figure 15. Check that the text in Figure 15 is not using a different font.

Author Response

The manuscript is interesting and deals with an important topic, especially in the face of increasing climate change. The editorial staff of the article could have been more careful. The necessary fixes are listed below:

Authors’ response: Thanks for your positive comment.

  1. The paper is more straightforward to read if each table and figure are immediately preceded by a reference in the text. If possible, please include the designation in the text before the table and the drawing. I mean Figure 1, Figure 2, Figure 7, Figure 14, Figure 15, Table 1 and Table 2.

Authors’ response: Thanks for making this good suggestion. The designation in the text before the tables and the figures have been added.

  1. If possible, please move the name of table 1 (line 118) to the next page.

Authors’ response: Done

  1. Does the table formatting comply with the publisher's recommendations? Please, check the Instructions for Authors and use the Microsoft Word template.

Authors’ response: Thanks for making this good suggestion. All tables are added in Microsoft Word template.

  1. Please describe in points the tasks listed in lines 140-146. It will be much more readable.

Authors’ response: Thanks for your positive comment. Three tasks are listed.

  1. There is no reference in the text to  Figure 6.

Authors’ response: Figure 6 is now discussed.

  1. If you describe the parameter designations (equation 1), use a colon after the "where" (line 197).

Authors’ response: Line 219: colon added.

  1. Do not use italic type in equation 2 (line 252) if equation 1 does not have italics (line 195).

Authors’ response: Line 279: Italic font is removed.

  1. If possible, move the description of Figure 10 so that it is on the same page as the Figure 10.

Authors’ response: Done

  1. If possible, move the description of Table 5 so that it is on the same page as the Table

Authors’ response: Done

  1. In Figure 12, the name "Soli" should not contain capital letters. In the remaining drawings, only the first letter of the description is capital.

Authors’ response: Done

  1. If possible, move the name of section 5.7 to the next page (line 347).

Authors’ response: Done

  1. Please add horizontal extension lines in Figure 15. Check that the text in Figure 15 is not using a different font.

Authors’ response: Done

Reviewer 2 Report

Formal errors:

1.       Please, keep and follow the rule for abbreviations - at the first occurrence, the abbreviation must be written out in full (NSW – New South Wales? – line 83, CBD, line 103, MCDA, WLC appears 1st time on line 128, explanation is on line 244 etc., similar SRTM etc.). However it is a common abbreviation, I recommend to explain also the abbreviation RWH (e.g. the GIS abbreviation is explained as well). Some parts of the text are (from my point of view) only hardly readable due to the large number of abbreviations.

2.       The rainfall stations on the Fig. 2 shall be marked with the rainfall station number, so there will be an interconnection of the Fig. 2 and Tab.1

3.       Fig. 5 – are there areas with the elevation of -31 m (below the sea level)? Presumably no, so please correct the legend range (min-max).

4.       Line 299 – “Fig. 9 … As shown below, a stream order layer value is ranging from 1 to 7.” On the Fig. 9, there are only 6 stream orders.

Logical errors, comments, recommendations:

1.       Line 84 – “The final suitability map depicts the spatial distribution of area with limited access to water…” How is the limited water access taken into account in your study? I regard this as a crucial point because there is no need (so far) to implement the RWH, if there are sufficient and accessible other water resources.

2.       Also, very important is the issue of the water quality, on the line 86 is “...RWH approach to reduce household water usage.” Which kind of “household” usage? Rainwater can be used for some activities in the households (e.g. gardening, maybe toilet flushing), but use it for other purposes (e.g. drinking water, hygiene, washing, kitchen) can be very problematic and economically ineffective. As found in later text (line 215-216) you want also to use the water for livestock and human use – this is somewhat confusing.

3.       Please define clearly in the paper, for what type of water use and consequently for which type of devices for RWH your study applies. In your paper you use just the general term RWH, but most of your selection criteria are not relevant for some common types of RWH, e.g. for the RWH from roofs (typical for households, see l. 86). For such methods of the RWH are all of your selection criteria (except land use, maybe rainfall) not relevant.

4.       Following on the previous point I recommend to strictly split the RWH methods according the planned rainwater use. As stated in previous points, the domestic, agricultural and industrial water use has totally different requirements on water quality and the RWH methods should be adapted on this requirements. Or do you mean to use the rainwater as a service / irrigation water in all three sectors? If so, this should be clearly stated and explained in the paper.

5.       Line 165 “Because a substantial volume of water needs to be stored….” How do you know, what volume of the water needs to be stored? How do you define “substantial volume”?

6.       Line 166 - What type of “storage structures” do you mean? Small dams, rain tanks, water reservoirs or…? This comment applies to the entire text of the paper.

7.       Line 190 – “Increased drainage density results in more runoff being produced [42]” Yes, it is a citation, but it sounds very strange to me. Where in the paper [42] is this citation? (I did not found it there) Or it is just your interpretation of some facts in this paper? BTW, the methodology in this paper is very similar to yours... just pls. note, that the paper [42] is dealing strictly with the RWH structures, defined as “….dam or water storage structures for conserving rainwater for irrigation.”

8.       Line 215-216 – “water for livestock and human use” I am not sure, but presumably you assume to store the rainwater in small dams (reservoirs) - how do you expect to ensure the water quantity and quality? The water for human use (drinking water) needs to be safe, i. e. it needs a water treatment, incl. disinfection etc.

9.       Line 240 – what do you mean by “socioeconomic criteria”? I am really not sure, if this can be evaluated by the distance to roads and settlements. BTW, if you harvest the rainwater in area with very high suitability (Fig. 13), then you have to transport the harvested water to the settlements/industry (presumably close to the roads) – so more than 9 kilometres?

10.   Line 252 – which values of the relative importance weights you used for your study? Please document this in the text / table.

The basic lack is the missing focus to a specific water use (e.g. agricultural use - irrigation) and consequent definition of the rainwater harvesting methods (structures). This leads also to the selection of criteria, methods, suitability etc. The current text and focus of the paper is very general, which leads to doubts about the used methods, criteria and of course the results.

Author Response

Formal errors:

  1. Please, keep and follow the rule for abbreviations - at the first occurrence, the abbreviation must be written out in full (NSW – New South Wales? – line 83, CBD, line 103, MCDA, WLC appears 1sttime on line 128, explanation is on line 244 etc., similar SRTM etc.). However it is a common abbreviation, I recommend to explain also the abbreviation RWH (e.g. the GIS abbreviation is explained as well). Some parts of the text are (from my point of view) only hardly readable due to the large number of abbreviations.

Authors’ response: All abbreviations are written in full form as appropriate.

  1. The rainfall stations on the Fig. 2 shall be marked with the rainfall station number, so there will be an interconnection of the Fig. 2 and Tab.1

Authors’ response: Done

  1. Fig. 5 – are there areas with the elevation of -31 m (below the sea level)? Presumably no, so please correct the legend range (min-max).

Authors’ response: DEM height range adjusted in Figure 5.

  1. Line 299 – “Fig. 9 … As shown below, a stream order layer value is ranging from 1 to 7.” On the Fig. 9, there are only 6 stream orders.

Authors’ response: Corrected: 1 to 6

Logical errors, comments, recommendations:

  1. Line 84 – “The final suitability map depicts the spatial distribution of area with limited access to water…” How is the limited water access taken into account in your study? I regard this as a crucial point because there is no need (so far) to implement the RWH, if there are sufficient and accessible other water resources.

Authors’ response: Thanks for the suggestion. The sentence is rewritten as “The final suitability map depicts the spatial distribution of areas suitable for using RWH to meet irrigation demand.”

  1. Also, very important is the issue of the water quality, on the line 86 is “...RWH approach to reduce household water usage.” Which kind of “household” usage? Rainwater can be used for some activities in the households (e.g. gardening, maybe toilet flushing), but use it for other purposes (e.g. drinking water, hygiene, washing, kitchen) can be very problematic and economically ineffective. As found in later text (line 215-216) you want also to use the water for livestock and human use – this is somewhat confusing.

Authors’ response: Thanks for your suggestion. Household water use is changed to “irrigation water use”. Also, “water for livestock and human use” is changed to “irrigation use”.

  1.      Please define clearly in the paper, for what type of water use and consequently for which type of devices for RWH your study applies. In your paper you use just the general term RWH, but most of your selection criteria are not relevant for some common types of RWH, e.g. for the RWH from roofs (typical for households, see l. 86). For such methods of the RWH are all of your selection criteria (except land use, maybe rainfall) not relevant. 

Authors’ response: We have specified the water use as “irrigation use” only.

  1. Following on the previous point I recommend to strictly split the RWH methods according the planned rainwater use. As stated in previous points, the domestic, agricultural and industrial water use has totally different requirements on water quality and the RWH methods should be adapted on this requirements. Or do you mean to use the rainwater as a service / irrigation water in all three sectors? If so, this should be clearly stated and explained in the paper.

Authors’ response: Thanks for the comment and suggestion. We have changed the water use to only “irrigation use”. We have changed the paper title as well to reflect this.

  1. Line 165 “Because a substantial volume of water needs to be stored….” How do you know, what volume of the water needs to be stored? How do you define “substantial volume”?

Authors’ response: The sentence is revised as below: “If a large volume of water is needed to meet irrigation demand, high slope areas may not be suitable as the storage structure could be more expensive in such areas.”

  1. Line 166 - What type of “storage structures” do you mean? Small dams, rain tanks, water reservoirs or…? This comment applies to the entire text of the paper. 

Authors’ response: The sentence is revised as below: “If a large volume of water is needed to meet irrigation demand, high slope areas may not be suitable as the storage structure (such as farm dam, check dam and contour bund) could be more expensive in such cases. Areas with a medium to low slope are more practical to reduce capital cost of storage structure.

  1. Line 190 – “Increased drainage density results in more runoff being produced [42]” Yes, it is a citation, but it sounds very strange to me. Where in the paper [42] is this citation? (I did not found it there) Or it is just your interpretation of some facts in this paper? BTW, the methodology in this paper is very similar to yours... just pls. note, that the paper [42] is dealing strictly with the RWH structures, defined as “….dam or water storage structures for conserving rainwater for irrigation.”

Authors’ response: Correct reference is added and the sentence is modified as below: “The volume of runoff loss via infiltration depends on drainage density and the higher the drainage density, the higher the RWH potential [40].”

  1. Line 215-216 – “water for livestock and human use” I am not sure, but presumably you assume to store the rainwater in small dams (reservoirs) - how do you expect to ensure the water quantity and quality? The water for human use (drinking water) needs to be safe, i. e. it needs a water treatment, incl. disinfection etc.

Authors’ response: “water for livestock and human use” is changed to “irrigation use”.

  1. Line 240 – what do you mean by “socioeconomic criteria”? I am really not sure, if this can be evaluated by the distance to roads and settlements. BTW, if you harvest the rainwater in area with very high suitability (Fig. 13), then you have to transport the harvested water to the settlements/industry (presumably close to the roads) – so more than 9 kilometres?

Authors’ response: Thanks for the suggestion. The following sentence is added: “It should be noted that the harvested rainwater should be used locally for irrigation to avoid high water transportation cost”.

  1. Line 252 – which values of the relative importance weights you used for your study? Please document this in the text / table.

Authors’ response: Weighted linear combination approach was applied in this study using the ArcGIS software after determining the normal weight of each layer and sub-layer however the weights are not saved in a table.

The basic lack is the missing focus to a specific water use (e.g. agricultural use - irrigation) and consequent definition of the rainwater harvesting methods (structures). This leads also to the selection of criteria, methods, suitability etc. The current text and focus of the paper is very general, which leads to doubts about the used methods, criteria and of course the results.

Authors’ response: Thanks for your constructive suggestion. The water us is specified as “irrigation use” and the water storage structures are specified (such as farm dam, check dam and contour bund).

Round 2

Reviewer 1 Report

Thank you for the changes to the manuscript so far. Please take note of my 3 more comments.

If you use the acronym RWH, it makes sense to explain it the first time you use it in the text. In line 17, I propose to add an abbreviation in parentheses. (Rainwater harvesting)

In Figure 12, the name "Soli" should not contain capital letters, if in the remaining drawings, only the first letter of the description is capital.

Please add horizontal extension lines in Figure 15. 

Author Response

Thank you for the changes to the manuscript so far. Please take note of my 3 more comments.

If you use the acronym RWH, it makes sense to explain it the first time you use it in the text. In line 17, I propose to add an abbreviation in parentheses. (Rainwater harvesting)

Authors’ response: Done

In Figure 12, the name "Soli" should not contain capital letters, if in the remaining drawings, only the first letter of the description is capital.

Authors’ response: Done, new figure is added.

Please add horizontal extension lines in Figure 15. 

Authors’ response: Figure 15 is redone.

Reviewer 2 Report

Formal errors:

1. OK

2. I see no changes in the Fig. 2

3. OK

4. OK

Which changes were made in all Tabs in the paper? I see no differences, maybe just some formatting of the tables.

Logical errors, comments, recommendations:

1. OK

2. OK

3. OK

4. OK

5. OK, see also the next point

6. There is too much uncertainty in this sentence - “if…may be …. could …” ….but OK

7. Well….. the exact citation of this source is “the lower the drainage frequency density, the lower the RWH potential” Your citation is not in contradiction, but if you cite an author, the citation must be verbatim or very precise.

8. OK

9. Once again – can the “socioeconomic criteria” be evaluated by the distance to roads and settlements? Please explain the causality and relationships of this evaluation in the paper.

10. I understand that you used the “weighted linear combination approach”, but my question was aimed to document the used VALUES of the particular weights in the paper (i.e. to document, which factors you considered as the most or less important ones).

Author Response

Formal errors:

  1. OK
  2. I see no changes in the Fig. 2

Authors’ response: Done

  1. OK
  2. OK

Which changes were made in all Tabs in the paper? I see no differences, maybe just some formatting of the tables.

Authors’ response: All tables are now in Microsoft word editable format as per journal guidelines.

Logical errors, comments, recommendations:

  1. OK
  2. OK
  3. OK
  4. OK
  5. OK, see also the next point
  6. There is too much uncertainty in this sentence - “if…may be …. could …” ….but OK

Authors response: These are rectified as much as possible.

  1. Well….. the exact citation of this source is “the lower the drainage frequency density, the lower the RWH potential” Your citation is not in s, but if you cite an author, the citation must be verbatim or very precise.

Authors’ response: Many thanks for the valuable feedback. The sentence is modified as below: “The volume of runoff loss via infiltration depends on drainage density and the lower the drainage density, the lower the RWH potential [40].”

  1. OK
  2. Once again – can the “socioeconomic criteria” be evaluated by the distance to roads and settlements? Please explain the causality and relationships of this evaluation in the paper.

Authors’ response: The following sentence is added: “Although socioeconomic status is not directly related to the proximity of roads and settlements, in rural areas, developments tend to happen near road network.”  

  1. I understand that you used the “weighted linear combination approach”, but my question was aimed to document the used VALUES of the particular weights in the paper (i.e. to document, which factors you considered as the most or less important ones).

Authors’ response: The following sentence is added:

“Based upon findings and evaluations of previous studies, the following weight values were used in this study: Rainfall: 20; Soil: 20, LULC: 20; Drainage Density: 20, Slope: 10, Stream: 5, Distance to Roads and settlement: 5.”